# Dietary Protein Intake and Physical Function in Māori and Non-Māori Adults of Advanced Age in New Zealand: LiLACS NZ

**DOI:** 10.3390/nu15071664

**Published:** 2023-03-29

**Authors:** Maia Lingman, Ngaire Kerse, Marama Muru-Lanning, Ruth Teh

**Affiliations:** 1Te Whatu Ora, Waitematā, Auckland 0622, New Zealand; 2School of Population Health, The University of Auckland, Auckland 1023, New Zealand; 3James Henare Māori Research Centre, The University of Auckland, Auckland 1023, New Zealand

**Keywords:** protein, physical performance, older adults, nonagenarians, indigenous

## Abstract

The population of older adults is growing exponentially. Research shows that current protein intake recommendations are unlikely to meet the ageing requirements and may be linked to reduced physical function. Ensuring optimal function levels is crucial for independence and quality of life in older age. This study aims to quantify the protein intake in those over 90 years of age and determine the association between historical protein intake (2011) and subsequent physical function at ten years follow-up (2021). Eighty-one participants (23 Māori and 54 non-Māori) undertook dietary assessment 24 h multiple-pass recall (MPR) and a standardised health and social questionnaire with physical assessment in 2011 and 2021. Intake24, a virtual 24 h MPR, was utilised to analyse dietary intake. Functional status was measured using the Nottingham Extended Activities of Daily Living Scale (NEADL), and physical performance was the Short Physical Performance Battery (SPPB). Māori men and women consumed less protein (g/day) in 2021 than in 2011 (P = 0.043 in men), but weight-adjusted protein intake in Māori participants over the ten years was not significantly reduced. Both non-Māori men and women consumed significantly less protein (g/day) between 2011 and 2021 (*p* = 0.006 and *p* = 0.001, respectively), which was also significant when protein intake was adjusted for weight in non-Māori women (*p* = 0.01). Weight-adjusted protein intake in 2011 was independently associated with functional status (NEADL score) in 2021 (*p* =< 0.001). There was no association between past protein intake and SPPB score (*p* = 0.993). Animal protein was replaced with plant-based protein over time. In conclusion, a reduction in protein intake was seen in all participants. The independent association between past protein intake and future functional status supports recommendations to keep protein intake high in advanced age.

## 1. Introduction

New Zealand has a growing population of older adults, with people aged over 85 years predicted to grow from 2% of the total population in 2020 to 4–5% in 2048 and 5–8% in 2073 [1]. The older Māori population is projected to grow more quickly [2]. Optimal quality of life is the priority as the population ages. Social relationships, environmental spaces, emotional well-being, health and functional ability are crucial factors impacting quality of life of older adults worldwide [3,4,5]. Both health and decline in functional ability are affected by ageing [6], and diet contributes most significantly to health losses from disease in New Zealand [7].

The New Zealand Adult Nutrition Survey 2008/09 reported that those over 71 years of age had the highest prevalence of inadequate protein intake (13.4% in men and 15.5% in women) than any other age group [8]. Adequate protein and energy intake are important for optimal health, particularly older adults. Adequate protein intake is essential in lean muscle mass maintenance [6,9,10]. Protein-energy malnutrition is prevalent in the institutionalised, hospitalised and up to 25% of community-dwelling populations of older people [11]. In community-dwelling older adults, mean age between 69 and 86 years, up to 45% of sample had protein intake below the current recommended protein intake (<0.8 g/kg BW/d) [12]. It is essential to achieve adequate protein intake, as prolonged protein-energy malnutrition can severely affect well-being, health and functional status [13].

Additionally, international researchers and nutrition agencies such as ESPEN and PROT-AGE suggest that current recommended protein intakes are insufficient to meet the requirements to maintain muscle strength and function [14,15,16]. Updated information on protein intake for older adults related to supporting nutritional status and health in ageing is needed.

Impaired mobility, functional decline and mortality are linked to reducing muscle mass and strength with increasing age, significantly impacting quality of life [17,18,19,20]. Adequate protein intake is associated with better maintenance of lean muscle mass and physical function in older adults [6,9,15,21]. Good physical function is essential not only on a micro-environmental level and meso-environmental level, affecting both an individual’s and the broader age groups collective quality of life, but also on a macro-environmental level. The requirement for support services due to age-related reduction in functional status and independence cost New Zealand Government $983 million in 2015 [22].

Investigating elements that may impact functional decline in ageing, including diet, may lead to strategies to offset the high cost to individuals and the government of providing support. The ageing demographic, with greater proportional increases in Māori, indigenous to NZ, justifies further investigation into the relationship between physical function and dietary protein intake in New Zealand [6,23]. Whilst frailty in this population has previously been investigated [24], there is little research on protein intake and physical function longitudinally in the 85+ demographic. Focusing research efforts on New Zealand’s older-aged population has the potential to positively impact the quality of life and health of this age group.

The aim of this study was to investigate the protein intake of over 90-year-olds in New Zealand and to measure the impact of dietary protein intake on functional status in Māori and non-Māori over age 90-year in New Zealand.

## 2. Materials and Methods

### 2.1. Life and Living in Advanced Age: A Cohort Study in New Zealand (LiLACS-NZ)

Te Puā waitanga o Ngā Tapuwae Kia ora Tonu, Life and Living in Advanced Age: A Cohort Study in New Zealand (LiLACS-NZ) is a cohort study with inception in 2010 investigating the predictors of successful ageing in adults of advanced age. This research was carried out within the Bay of Plenty District Health Board and Lakes District Health Board areas. The 10th year follow up was completed in 2021 with participants over 90 years of age. A detailed study protocol has been previously published [25], and data outcomes have been published over the follow-up period [26,27,28,29,30,31,32,33,34,35,36,37,38]. In brief, Māori, indigenous peoples of Aotearoa New Zealand, aged 80–90 years, (10-year birth cohort 1920–1930) and non-Māori aged 85 years (single birth year cohort, 1925) were identified through multiple overlapping strategies, invited, given informed consent, and then interviewed in 2010–2011. A wider age band was necessary for Māori to allow Māori-specific analyses and provide pro-equity data analysis. Participants were followed up yearly for five years with one 10-year follow-up contact in 2021 (delayed by COVID restrictions from the planned year 2020). A detailed nutrition assessment completed in 2011 forms the baseline for this paper [39]. At baseline, two dietary assessments (24 h multiple pass recall) and a health and well-being questionnaire with physical assessment were conducted by trained interviewers. One 24 h multiple pass recall and the health and well-being interview were repeated for the 10-year follow-up of the study (2021; wave seven). Written informed consent was gained from all participants in this study.

### 2.2. Dietary Assessment: Multiple-Pass 24-H Recall (MPR)

Trained interviewers conducted a multiple-pass 24 h recall. At 10-year follow-up, this was completed on one occasion to reduce participant burden. The MPR protocol has been proven suitable for the oldest-old population and matches the dietary assessment methods used in the LILACS-NZ and Newcastle 85+ studies [39].

This study utilised an online version of the MPR, the Intake24 virtual dietary assessment system developed by Newcastle University, UK [40,41] with incorporated data of common foods from the New Zealand FOODfiles 2016 database (e.g., mussels, pipi, pūha, silverbeet) [42]. INTAKE24 has a built-in photographic atlas and uses household measures to aid portion size estimations. Reported foods that were not already listed in the Intake24 were identified in the dataset, and the closest alternatives were found in the FOODfiles 2016 database or nutrition information panels (NIPs) by nutrition trained researchers. Nutrient analysis for these items was manually entered into the dataset. Interviewers took care to ensure accurate estimates were attained to ensure optimal accuracy in reporting nutrient intake. Missing foods were coded into the most appropriate food group categories.

For participants residing in rest homes who may have memory or cognitive decline and are not self-preparing meals, food service managers were approached for information regarding the menu and recipes used in rest homes where able; otherwise, rest home managers or nurse managers were approached. Nurses and allied health assistants gave information regarding patient intake as they were directly involved in mealtimes. They were asked to note how much participants left on their plates the day prior to interviewing. A copy of the recipes was also obtained. A short questionnaire collected demographic information from all participants, assessed whether the reported intake was usual, perceived ability to eat a healthy diet, weekly expenses on food and drink, and noted issues with the interview and whether the interviewer thought this recall reflected the participants’ true intake.

#### Selection of Nutrients

INTAKE24 data were collated and analysed using FOODfiles 2016. Key macronutrients analysed included energy (Kcal), protein (gram), carbohydrate (g), total fat (g), saturated fat (g) and cholesterol (g).

### 2.3. Health and Well-Being Questionnaire and Physical Assessment

Trained interviewers conducted a standardised structured Health and Well-being Questionnaire and Physical Assessment. The development of these interviews and examinations are described [25]. Relevant to this paper, data collected were age, marital status, living situation, anthropometric data (height, weight, body mass index (BMI)), body composition using a Tanita Bioimpedance Analysis BC545 (BIA) scale (total body fat, bone mass, muscle mass), sitting and standing blood pressure, grip strength using a handheld dynamometer-Grip D, the Physical Activity Scale (PASE) [43], Nottingham Extended Activities of Daily Living Scale (NEADL) [44] and Short Physical Performance Battery (SPPB) [45]. Demographic information (sex, ethnicity, education, and NZ Deprivation Index) was collected once at baseline (2010).

Physical assessments were not completed if it was unsafe for the participant or if they opted out of this part of the assessment. Reasons for incomplete physical assessment were recorded.

### 2.4. Statistical Analysis

Analyses were completed separately for Māori and non-Māori where possible. Māori have different culture-related foods [46,47] and are disproportionately socio-economically disadvantaged than other ethnic groups in New Zealand. Separate analyses were completed to inform protein intake and the potential impact on physical function in Māori living into their ninth decades.

Descriptive analyses were completed for all variables. Continuous data were presented as means and standard deviation (SD) or median, interquartile range (IQR), and categorical data such as number (*n*) and percentage (%).

Further computation on dietary protein intake data was completed to reflect current New Zealand Nutrient Reference Values: Recommended Daily Intake (RDI); inadequate protein intake (<0.8 g/kg BW/day), adequate protein intake (>0.8 g/kg BW/day) and Estimated Average Requirement (EAR) (>0.86 g/kg BW/day in men and >0.75 g/kg BW/day in women) [48]. Sources of protein were reported according the 2008/09 New Zealand Adult Nutrition Survey NZ food groups.

Paired *t*-test (or McNemar) was used to determine difference in variables of interest between 2011 and 2021 (see Table 1 and Table 2 in the results section). Independent *t*-tests (or Kruskal–Wallis tests) were used to determine the association between protein intake (g/day, g/kg BW/day) and gender. Pearson’s (or Spearman’s) correlation test was used to determine the association between protein intake and functional status (NEADL), physical performance (SPPB), BIA measures and PASE score. All results for univariate analysis were presented in mean (SD) or median (IQR) and P-value. Variables with *p* < 0.2 were included in the multiple regression model. Generalised linear models (GLM) were completed to determine the independent association between protein intake (g/Kg BW/day) and physical function as the dependent variable, adjusting for relevant confounders (age, gender, ethnicity, education status, socio-economic status, body weight, BMI, muscle mass, energy intake, gait speed, physical activity levels and living situation). Three models were tested, each with increasing number of possible confounding variables, and the final models are presented in the main text (see Table 3 and Table 4 in the results section). Models 1 and 2 can be found in the Appendix A. We did not split the data by ethnicity in the final GLM because there were too few Māori men to provide stable models. Beta coefficients, 95% confidence intervals and P-values were presented. The significance level is set at *p* < 0.05. Data were analysed using the statistical package SPSS version 27.0 (IBM SPSS Statistics for Windows).

## 3. Results

### 3.1. Participant Characteristics, Physical Function and Nutrient Intake

A total of eighty-one participants were recruited into the current study (23 Māori and 58 non-Māori). This was 9% (81/937) of those still alive in original LiLACS NZ and 72% (81/112) of those invited; 5 did complete the dietary assessment interview. Reasons for nonresponse: decline or unable to consent (21), and not contactable (5). Participant characteristics, physical function measures and nutrient intake are presented for Māori and non-Māori participants separately in Table 1 and Table 2, respectively, showing changes in measures from 2011 to 2021.

#### 3.1.1. Māori Participants

There were five Māori men and 18 Māori women aged between 92 and 94 years old. Fifty-six percent of Māori women and all men lived with others. Over the ten years, SPPB and NEADL scores decreased in both men and women. A similar trend was observed in grip strength and physical activity (PASE) score, and this was only significant in women (Table 1).

Overall, the total energy intake significantly reduced to 1054 kcal/day and 1337 kcal/day in men and women, respectively, between 2011 and 2021. Protein intake almost halved in Māori men from 81.2 g/day to 40.8 g/day (*p* = 0.043) but was relatively stable in Māori women (from 56 g/day to 51.7 g/day, *p* = 0.096). The mean weight-adjusted protein intake remained stable in Māori women (0.86 g/kg BW/day) but declined in Māori men (0.69 g/kg BW/day), although it was not statistically significant. Percent of energy from protein remained relatively stable, about 16% for both men and women, meeting the lower end of the acceptable macronutrient distribution range (AMDR). However, the percentage of animal protein and plant-based protein has changed (Figure 1). There was a significant reduction in fat intake in both men (*p* = 0.043) and women (*p* = 0.009) between 2011 and 2021; a similar declining trend was also observed for cholesterol intake. However, the percentage of energy from fat did not change significantly. Interestingly, the percentage of energy from carbohydrates increased significantly in Māori women from 44.7% to 50.3% (*p* = 0.022). Figure 2 shows the primary dietary sources of protein in Māori in 2011 and 2021. A key difference between these two time periods was that poultry, beef and veal, and lamb and mutton did not make it to the ten most common dietary protein sources in 2021. Interestingly, fish and seafood are the most common source of dietary protein in Māori, which are followed closely by bread.

#### 3.1.2. Non-Māori Participants

The mean age of non-Māori participants was 95 years. The majority of men and women lived in private homes with others. All measures of physical function declined significantly over the ten years. The decline in mean grip strength was more notable in men (from 33.2 to 24.4 kg) than in women (17.8 to 15 kg). Notably, there was a significant decline in physical activity (PASE score from 116 to 27 and 88 to 28 in men and women, respectively) (Table 2).

The overall energy intake reduced from 2040 to 1581 kcal in men (*p* = 0.003) and 1637 to 1477.6 kcal in women between 2011 and 2021. Mean protein intake (g/day) and body weight-adjusted protein intake declined in both men and women between 2011 and 2021; body weight-adjusted protein intake in women declining significantly from 1.00 ± 0.29 g/kg BW/day to 0.84 g/kg BW/day (*p* = 0.010), but this was not significant in men (1.08 to 0.94 g/kg BW/day, *p* = 0.295). Non-Māori men had a significant reduction in carbohydrate (*p* = 0.025) and fat intake (*p* = 0.004) but not women. Non-Māori women had a significant reduction in percent energy intake from protein from 16.4% in 2011 to 14.2% in 2021 (*p* = 0.001); this was also reflected in a significant decline in the proportion of the group not attaining the AMRD for protein; this was not observed in their male counterparts (Table 2). We observed that the distribution of animal versus plant-based protein was changed from 60% animal protein to 40% plant-based protein in 2011 to 52%:48% animal: plant-based protein (Figure 1). There was a significant increase in percent energy from carbohydrate between 2011 and 2021 (*p* < 0.05) and only a modest reduction in percent energy from fat (*p* > 0.05). Figure 3 shows the primary sources of protein in non-Māori in 2011 and 2021; dairy products, non-alcoholic beverages (e.g., hot drinks and fruit juice) and biscuits replaced milk, fish and seafood, and cheese as the top ten food groups of dietary protein in 2021.

#### 3.1.3. Relationship between Protein Intake in 2011 and Physical Function Measures in 2021

The changes in function levels (NEADL and SPPB score) between 2011 and 2021 in relation to achieving the Recommended Daily Intake (RDI) for protein (≥0.8 g/kg BW/day) in 2011 by sex and ethnicity are presented visually in Appendix A). Participants with inadequate protein intake in 2011 had a larger reduction in NEADL score over the 10-year period than those with adequate intake (38% and 43% reduction in those who did and did not meet the RDI in 2011, respectively). Participants with inadequate protein intake in 2011 had a mean reduction in SPPB score of 51% over ten years compared to 46% in those with adequate protein intake.

Regression models examined the relationship between physical function outcomes (NEADL and SPPB scores in 2021) and protein intake in 2011. The models were not split by ethnic groups due to the small sample size. Models 1 and 2 can be found in the Appendix A. The multicollinearity between independent variables was examined, and we found that historical NEADL was correlated with current PASE. We chose to include PASE over NEADL in the final GLM because it fit the model better. Table 3 shows the final regression model of the relationship between weight-adjusted protein intake in 2011 and 2021 NEADL score. We observed that weight-adjusted protein intake in 2011 was independently associated with 2021 NEADL score (13.6 (95% CI 6.7–20.6), *p* < 0.001) but not SPPB score (*p* = 0.99) (Table 4). Male sex was associated with lower SPPB score at 10 years follow-up (*p* = 0.012); and those living in low NZ deprivation areas compared to high NZ deprivation areas was associated with higher a SPPB score at 10 years follow-up (*p* = 0.048).

**Table 1 nutrients-15-01664-t001:** Demographic Characteristics, Physical Function and Nutrient Intake in Māori Participants.

		Men, *n* = 5			Women, *n* = 18		
Age, years (mean ± SD)		93.5 ± 3.8			92.0 ± 2.4		
Education, *n* (%)	Primary or no school	2 (40.0)			3 (16.7)		
	Secondary school, no qualification	1 (20.0)			9 (50.0)		
	Secondary school, qualification	2 (40.0)			4 (22.2)		
	Trade/tertiary qualification	0 (0.0)			2 (11.1)		
Marital status, *n* (%)	Never married	1 (33.3)			2 (18.2)		
	Married	2 (66.7)			6 (54.5)		
	Widow	0 (0.0)			1 (9.1)		
	Divorced	0 (0.0)			2 (18.2)		
NZ deprivation index, *n* (%) ^a^	1–3 (low)	2 (40.0)			2 (11.1)		
	4–7 (medium)	1 (20.0)			5 (27.8)		
	8 and above (high)	2 (40.0)			11 (61.1)		
Who do you live with? *n* (%)	Alone	0 (0.0)			8 (44.4)		
	With others	5 (100.0)			10 (55.6)		
Where do you live? *n* (%)	Private home	4 (100.0)			12 (70.6)		
	Retirement village	0 (0.0)			2 (11.8)		
	Rest home	0 (0.0)			3 (17.6)		
	Other	0 (0.0)			0 (0.0)		
		2011	2021	*p*-Value	2011	2021	*p*-Value
BMI (Kg/m^2^), mean ± SD		28.9 ± 5.4	29.0 ± 4.7	0.655	28.0 ± 3.4	24.5 ± 5.0	0.161
Height (m), mean ± SD		1.65 ± 0.01	1.61 ± 0.1	0.180	1.56 ± 0.1	1.54 ± 0.1	0.017 *
Weight (Kg), mean ± SD		78.0 ± 13.4	70.9 ± 8.9	0.285	67.7 ± 7.9	58.1 ± 9.1	0.002 *
Muscle mass (Kg), mean ± SD		52.4 ± 3.4	54.9 ± 2.5	0.285	39.1 ± 3.0	35.9 ± 2.4	0.213
Fat mass (%), mean ± SD		28.1 ± 9.5	19.7 ± 7.8	0.109	38.2 ± 4.6	33.9 ± 5.7	0.025 *
Bone mass (Kg), mean ± SD		2.8 ± 0.2	2.5 ± 0.2	0.276	2.1 ± 0.2	2.0 ± 0.2	0.048 *
Systolic BP standing (mmHg), mean ± SD		147 ± 11	161 ± 12	0.285	150 ± 19	154 ± 13	0.398
Systolic BP sitting (mmHg), mean ± SD		151 ± 22	160 ± 28	0.715	148 ± 14	153 ± 17	0.477
Diastolic BP standing (mmHg), mean ± SD		85 ± 17	80 ± 9	0.109	92 ± 11	88 ± 10	0.018 *
Diastolic BP sitting (mmHg), mean ± SD		77 ± 7	72 ± 12	0.715	83 ± 10	85 ± 9	0.929
NEADL score, median (IQR)		17.5 (16.0–18.0)	16.5 (14.0–19.0)	1.000	19.5 (18.0–21.0)	15.0 (8.0–20.0)	<0.001 *
SPPB score, mean ± SD		9.8 ± 1.7	4.5 ± 5.5	0.285	9.2 ± 3.4	6.8 ± 3.6	0.006 *
Grip strength (Kg), mean ± SD		33.0 ± 5.8	22.4 ± 6.1	0.109	21.4 ± 3.3	16.7 ± 4.1	0.004 *
Total PASE score, median (IQR)		145 (107–193)	32 (0–185)	0.285	133 (83–167)	70 (18-93)	0.013 *
Energy intake (kcal), mean ± SD		1874 ± 705	1054 ± 515	0.05 *	1483 ± 331	1337 ± 473	0.026 *
Protein intake (g), mean ± SD		81.2 ± 27.5	40.8 ± 17.62	0.043 *	56.0 ± 16.5	51.7 ± 22.3	0.096
Protein intake (g/Kg BW/day), mean ± SD		1.09 ± 0.44	0.69 ± 0.07	0.109	0.86 ± 0.26	0.86 ± 0.36	0.859
Carbohydrate intake (g), mean ± SD		194.7 ± 90.4	132.5 ± 76.5	0.138	166.2 ± 46.9	166.9 ± 67.6	0.433
Fat intake (g), mean ± SD		86.0 ± 37.6	40.3 ± 18.8	0.043 *	63.9 ± 16.0	48.1 ± 17.3	0.009 *
Saturated fat intake (g), mean ± SD		34.7 ± 15.2	16.1 ± 9.6	0.080	26.6 ± 8.3	21.7 ± 9.9	0.109
Cholesterol intake (g), mean ± SD		282.4 ± 171.9	178.3 ± 171.4	0.043 *	221.1 ± 110.6	127.9 ± 64.4	0.016 *
Percent energy intake from protein (%), mean ± SD *n* (%) within the AMDR (15–25% of energy intake)		17.7 ± 3.3 4 (80.0%)	15.9 ± 4.4 3 (60.0%)	0.686 1.000	15.2 ± 3.5 9 (64.3%)	15.5 ± 4.8 5 (35.7%)	0.826 0.219
Percent energy intake from carbohydrate (%), mean ± SD *n* (%) within the AMDR (45–65% of energy intake)		41.1 ± 12.4 2 (40%)	48.6 ± 5.1 3 (60%)	0.225 1.000	44.7 ± 6.1 7 (15%)	50.3 ± 9.6 8 (57.1%)	0.022 * 1.000
Percent energy intake from fat (%), mean ± SD *n* (%) within the AMDR (20–35% of energy intake)		41.7 ± 13.3 2 (40%)	35.7 ± 5.5 2 (40%)	0.225 1.000	39.0 ± 5.9 4 (28.6%)	32.7 ± 8.4 7 (50.0%)	0.056 0.453

^a^ NZ Deprivation Status was collected in 2010. Low: NZ Deprivation (Dep) Index 1–3, Middle: NZ Dep Index 4–7, High: NZ Dep Index 8–10. Notes: AMDR, Acceptable Macronutrient Distribution Range; BW, body weight; g, gram; Kg, kilogram; NEADL, Nottingham Extended Activities of Daily Living Scale; PASE, Physical Activity Scale SD, standard deviation; SPPB, Short Physical Performance Battery. * Paired *t*-test (or McNemar) was used to determine difference in variables of interest between 2011 and 2021, statistically significant at *p* < 0.05.

**Table 2 nutrients-15-01664-t002:** Demographic Characteristics, Physical Function and Nutrient Intake in Non-Māori Participants.

		Men, *n* = 26			Women, *n* = 32		
Age, years (mean ± SD)		95.2 ± 0.4			95.3 ± 0.4		
Education, *n* (%)	Primary or no school	3 (11.5)			1 (3.1)		
	Secondary school, no qualification	9 (34.6)			11 (34.4)		
	Secondary school, qualification	7 (26.9)			10 (31.3)		
	Trade/Tertiary qualification	7 (26.9)			10 (31.3)		
Marital status, *n* (%)	Never married	0 (0.0)			0 (0.0)		
	Married	6 (66.7)			1 (12.5)		
	Widow	3 (33.3)			5 (62.5)		
	Divorced	0 (0.0)			2 (25.0)		
NZ deprivation index, *n* (%) ^a^	1–3 (low)	8 (30.8)			5 (15.6)		
	4–7 (medium)	11 (42.3)			11 (34.4)		
	8 and above (high)	7 (26.9)			16 (50.0)		
Who do you live with? *n* (%)	Alone	8 (30.8)			14 (43.8)		
	With others	18 (69.2)			18 (56.3)		
Where do you live? *n* (%)	Private home	12 (52.2)			13 (41.9)		
	Retirement village	8 (34.8)			6 (19.4)		
	Rest home	3 (13.0)			12 (38.7)		
	Other	0 (0.0)			0 (0.0)		
		2011	2021	*p*-Value	2011	2021	*p*-Value
BMI (Kg/m^2^), mean ± SD		26.4 ± 3.6	24.7 ± 4.6	0.013 *	27.1 ± 3.9	26.0 ± 3.3	0.398
Height (m), mean ± SD		1.68 ± 0.1	1.64 ± 0.1	<0.001 *	1.56 ± 0.1	1.52 ± 0.1	<0.001 *
Weight (Kg), mean ± SD		74.2 ± 9.5	65.8 ± 11.4	0.001 *	65.8 ± 9.3	62.0 ± 9.4	0.002 *
Muscle mass (Kg), mean ± SD		50.96 ± 4.7	45.71 ± 3.4	0.034 *	38.03 ± 3.9	36.81 ± 3.4	0.266
Fat mass (%), mean ± SD		27.39 ± 6.5	24.97 ± 10.0	0.504	37.65 ± 7.2	35.23 ± 8.7	0.085 *
Bone mass (Kg), mean ± SD		2.71 ± 0.2	2.46 ± 0.2	0.003 *	2.06 ± 0.27	1.98 ± 0.2	0.005 *
Systolic BP standing (mmHg), mean ± SD		147 ± 19	142 ± 29	0.510	152 ± 22	148 ± 17	0.717
Systolic BP sitting (mmHg), mean ± SD		147 ± 20	139 ± 24	0.083	147 ± 21	144 ± 26	0.728
Diastolic BP standing (mmHg), mean ± SD		83 ± 10	74 ± 16	<0.001 *	87 ± 15	83 ± 1	<0.001 *
Diastolic BP sitting (mmHg), mean ± SD		79 ± 11	70 ± 13	0.011 *	77 ± 12	78 ± 15	0.687
NEADL score wave 7, median (IQR)		18.0 (15.0–19.0)	12.0 (8.0–16.0)	<0.001 *	20.0 (19.0–21.0)	10.0 (6.0–16.0)	<0.001 *
SPPB score wave 7, mean ± SD		10.4 ± 1.5	5.1 ± 3.4	<0.001 *	8.5 ± 2.5	4.7 ± 3.5	<0.001 *
Grip strength (Kg), mean ± SD		33.2 ± 5.8	24.4 ± 6.2	<0.001 *	17.8 ± 4.2	15.0 ± 3.61	<0.001 *
Total PASE score, median (IQR)		116 (86–151)	27 (0–67)	<0.001 *	88 (55–142)	28 (9–83)	<0.001 *
Energy (kcal), mean ± SD		2040 ± 636	1581 ± 464	0.003 *	1637 ± 418	1478 ± 475	0.086
Protein (g), mean ± SD		80.3 ± 20.7	61.2 ± 27.4	0.006 *	66.7 ± 18.6	51.8 ± 15.6	0.001 *
Protein (g/Kg BW/day), mean ± SD		1.08 ± 0.29	0.94 ± 0.38	0.295	1.00 ± 0.29	0.84 ± 0.26	0.010 *
Carbohydrate (g), mean ± SD		212.9 ± 61.5	187.2 ± 66.1	0.025 *	175.2 ± 44.6	180.7 ± 66.4	0.393
Fat (g), mean ± SD		86.2 ± 33.3	61.5 ± 29.9	0.004 *	71.6 ± 27.2	59.3 ± 30.9	0.106
Saturated fat (g), mean ± SD		35.5 ± 16.3	26.1 ± 14.2	0.023 *	28.3 ± 14.0	26.4 ± 13.5	0.572
Cholesterol (mg), mean ± SD		302.4 ± 169.0	222.2 ± 128.5	0.062	249.3 ± 127.9	239.9 ± 170.9	0.428
Percent energy intake from protein (%), mean ± SD *n* (%) within the AMDR (15–25% of energy intake)		16.1 ± 2.7 16 (61.5%)	15.9 ± 5.9 14 (53.8%)	0.849 0.791	16.4 ± 3.1 20 (66.7%)	14.2 ± 2.8 11 (36.7%)	0.001 * 0.035 *
Percent energy intake from carbohydrate (%), mean ± SD *n* (%) within the AMDR (45–65% of energy intake)		42.9 ± 8.9 7 (26.9%)	47.6 ± 10.5 12 (46.2%)	0.038 * 0.227	43.5 ± 7.6 14 (46.7%)	49.0 ± 10.8 18 (60.0%)	0.037 * 0.424
Percent energy intake from fat (%), mean ± SD *n* (%) with the AMDR (20–35% of energy intake)		37.4 ± 6.3 7 (26.9%)	34.0 ± 9.9 14 (53.8%)	0.052 0.118	38.6 ± 7.1 11 (36.7%)	35.6 ± 11.0 13 (43.3%)	0.271 0.804

^a^ NZ Deprivation Status was collected at in 2010. Low: NZ Deprivation (Dep) Index 1–3, Middle: NZ Dep Index 4–7, High: NZ Dep Index 8–10. Notes: AMDR, Acceptable Macronutrient Distribution Range; BW, body weight; g, gram; Kg, kilogram; NEADL, Nottingham Extended Activities of Daily Living Scale; PASE, Physical Activity Scale SD, standard deviation; SPPB, Short Physical Performance Battery. * Paired *t*-test (or McNemar) was used to determine difference in variables of interest between 2011 and 2021, statistically significant at *p* < 0.05.

**Table 3 nutrients-15-01664-t003:** Multivariate Regression Model Examining the Relationship between 2011 Protein Intake, as Predictor, and 2021 NEADL Score in all Participants ^#^.

Variables	B (95% CI), *p*-Value
2011 Protein intake (g/kg BW/day) ^#^	13.59 (6.67–20.51), <0.001 *
2011 Energy intake (kcal/day) ^#^	−0.009 (−0.015–−0.003), 0.004 *
Age, years	−0.32 (−0.87–0.23), 0.253
Gender (ref: Women)	0.49 (−2.77–3.76), 0.767
Ethnicity (ref: Māori)	−1.43 (−9.49–6.64), 0.729
NZ deprivation index ^a^ low (ref: High) medium (ref: High)	0.02 (−2.31–2.35), 0.988 −2.01 (−3.80–−0.22), 0.028 *
Current living arrangement ^#^ Who do you live with? (ref: with others)	2.83 (0.89–4.77), 0.004 *
Current housing situation ^#^ Where do you live? private house (ref: rest home) retirement village (ref: rest home)	2.76 (0.17–5.35), 0.037 * 3.82 (0.72–6.91), 0.016 *
Current ^#^ PASE ^b^ score	0.04 (0.03–0.05), <0.001 *
Current ^#^ Fat mass (%)	0.05 (−0.06–0.15), 0.385
Current ^#^ Grip strength (kg)	0.16 (−0.06–0.37), 0.151

^#^ Current data collected in 2021. * Generalised linear models (GLM) adjusted for relevant confounder, statistically significant at *p* < 0.05. ^a^ NZ Deprivation index: Low-status 1–3, Middle-status 4–7, High-status 8–10. ^b^ PASE, Physical Activity Scale for the Elderly. Note: Interaction term Ethnicity*Wave 2 Protein intake; *p* = 0.012; Interaction term Ethnicity×Wave 2 energy intake; *p* = 0.018. There is a observed correlation between wave two NEADL and wave 7 PASE (r = 0.356, *p* = 0.003). Due to the multicollinearity of this data, we chose to include PASE in the GLM as it fit the model better.

**Table 4 nutrients-15-01664-t004:** Multivariate Regression Model Examining the Relationship between 2011 Protein Intake, as Predictor, and 2021 SPPB Score in all Participants ^#^.

Variables	B (95% CI), *p*-Value
2011 Protein intake (g/kg BW/day) ^#^	−0.030 (−6.858–6.798), 0.993
2011 Energy intake (kcal/day) ^#^	−0.001 (−0.005–0.003), 0.597
Age, years	0.207 (−0.281–0.694), 0.406
Gender (ref: Women)	−4.675 (−8.324–−1.027), 0.012 *
Ethnicity (ref: Māori)	−5.841 (−12.703–1.021), 0.095
Education status (ref: tertiary qualification) no or primary school secondary school, no qualification secondary school, qualification Trade	−1.154 (−6.238–3.931), 0.657 −1.311 (−4.188–1.565), 0.371 1.338 (−1.274–3.950), 0.315 1.187 (−2.176–4.550), 0.489
NZ deprivation index (ref: High) ^a^ Low Medium	−1.813 (−3.607–−0.018), 0.048 * −0.703 (−2.406–1.000), 0.418
Current living arrangement^#^ Who do you live with (ref: with others)	−0.551 (−2.478–1.376), 0.575
Current housing situation^#^ Where do you live? private house (ref: rest home) retirement village (ref: rest home)	1.423 (−1.289–4.135), 0.304 0.945 (−1.873–3.763), 0.511
Current ^#^ PASE ^b^ score	0.001 (−0.001–0.014), 0.828
Current ^#^ Fat mass (%)	−0.086 (−0.169–−0.003), 0.043 *
Current ^#^ Grip strength (kg)	0.308 (0.059–0.558), 0.015 *
2011 SPPB score	0.299 (−0.122–0.719), 0.164

^#^ Current data collected in 2021. * Generalised linear models (GLM) adjusted for relevant confounder, statistically significant at *p* < 0.05. ^a^ NZ Deprivation index: Low status 1–3, Middle status 4–7, High status 8–10. ^b^ PASE, Physical Activity Scale for the Elderly. Note: No interaction between ethnicity and wave 2 protein or energy intake (Interaction terms; Ethnicity × * Wave 2 Protein intake, *p* = 0.848 and Ethnicity * Wave 2 Energy intake, *p* = 0.099).

## 4. Discussion

This study aims to investigate protein intake in New Zealand nonagenarians and determine the impact of dietary protein intake on functional status in Māori and non-Māori at ten years follow-up. We found that intake reduced over 10 years along with physical activity, functional status and physical performance. Higher weight-adjusted protein intake (g/kg BW/day) (2011) was associated with better subsequent functional status ten years later. This finding supports the hypothesis that protein intake can impact functional status in those of advanced age.

### 4.1. Macronutrient Intake

Reducing dietary intake is common with ageing [16,49]. We found reductions in protein and fat intake with an increase in percentage energy from carbohydrates over ten years follow-up of octogenarians. This may be attributed to known risk factors such as oral health [29,50] and food cost [29]. The ten-year interval between two dietary assessments data points limit further inference. Other unknown factors affecting dietary intake, such as food accessibility, food preparation, and attitude towards food intake, may influence food choices. Interestingly, fish and seafood are the primary protein sources for Māori participants, which are followed closely by bread. Although the number of people in this sample of older Māori was small, it does appear that Māori were more able to maintain dietary protein intake than non-Māori. This is probably linked to the living environment, where Māori live closer to the sea and the provision of seafood from family and whānau, as was observed in 2011 [29]. For non-Māori, bread was the main source of protein. We hypothesise that the participants in their nineties may have replaced protein and fat consumption with carbohydrate foods which are more accessible and easier to prepare.

#### Protein Intake

Older adults in New Zealand have the highest level of inadequate protein intake compared to any other age group [8]. Protein requirements are amplified with increasing age due to age-related physiological changes, changes in dentition, food preferences, health status and social circumstances [6,15,16,49,51,52]. In the New Zealand Adult Nutrition Survey (NZ ANS) 2008/09, Māori men and women over 51 consumed a mean of 95 g/day and 68 g/day of protein, respectively [8]. In our study, Māori men in their nineties consumed about 40 g of protein per day, and Māori women consumed about 52 g/day. Protein intake, when adjusted by body weight, declined significantly in Māori men (from 1.09 ± 0.44 to 0.69 ± 0.07g/kg BW/day) while it remained stable in Māori women (0.86 ± 0.36 g/kg BW/day). The marked difference in protein intake observed in Māori men needs to be interpreted with caution, as the small sample is likely to have skewed the value. Our study observed that while the percentage energy from protein is relatively stable over time, the protein sources switched from animal protein as the primary source in 2011 to plant-based protein in 2021. Future research is needed to confirm our findings.

Among non-Māori participants, reduction in protein intake was more significant in women than men. Protein intake adjusted by body weight for women reduced from 1.00 ± 0.29 g/kg BW/day in 2011 to 0.84 ± 0.26 in 2021, and percentage energy from protein reduced from 16% to 14%. Both measures of protein intake were relatively stable in men. In non-Māori, animal protein was a major source of protein in 2011 and 2021. However, the ratio between animal and plant-based protein was closer to 1 in 2021 than the 60:40 animal-protein ratio of 1.5 in 2011. Research that shows that plant-based proteins have lower leucine content, reduced digestibility, and are deficient in some essential amino acids which are needed for muscle anabolism compared to animal-based proteins. Although it has been postulated that this could be negated by increasing volume of plant-based protein consumed, further research is required [53,54].

Overall, the quantity and quality of protein intake changed over time and varied by sex in Māori and non-Māori. A consistent finding across both sexes in the sample, irrespectively of ethnic groups, was that in all participants, the recommended percentage energy from protein hovered around the lower range of the Acceptable Macronutrient Distribution Range (AMDR), i.e., 15–25% energy from protein [48]. Animal protein was replaced with plant-based protein over time. It is difficult to distinguish exactly what this may be related to, although it would be feasible to believe this may occur due to changes in dentition (ability to chew animal-based protein) taste change/food preference and anorexia of ageing, which has been discussed in the literature [16].

### 4.2. Relationship between Protein Intake and Physical Function

Physical function is associated with daily living activities [20,55]. We observed NEADL and SPPB scores decrease during the 10-year follow-up period. This is in line with previous research showing a decreasing trend in physical function in adults 65 years and older, and the annual decline rate was greater in the 85+ group compared to those aged 65 to 74 years [56]. Granic et al. reported an inverse association between age, grip strength and Timed Up and Go Test (TUG) over five years in the Newcastle 85+ study and that the declines were not different between those who had low or high protein intake at baseline (<1 g/kg aBW/d vs. ≥1 g/kg aBW/d) [57]. We reported previously that protein intake was not associated with grip strength over five years [30]. Our current study extends the evidence on the association between protein intake and subsequent physical function at ten years follow-up. We found a positive association between weight-adjusted protein intake at age 80-years+ with functional status (*p* < 0.001) over 10 years but not with physical performance (*p* = 0.094). Functional status is correlated with physical performance, but it is a different measure impacted by psychological, environmental and social factors [58,59,60]. This finding supports the hypothesis that protein intake is associated with future functional status. The potential mechanism by which of protein intake benefits function may not only be through physical performance or muscle strength. Other age-related physiological changes throughout the life course may have a role.

The physical activity level (PASE score) of participants in the current study also decreased dramatically in all participants over the ten years (from a median of 93–150 in wave two to 27–70 in wave seven).

### 4.3. Relationship between Socio-Demographic Characteristics, Physical Activity and Physical Function

Physical activity levels usually are associated with higher protein intake and improved physical performance and ability to perform activities of daily living [6,61,62], so it is not surprising that we saw a reduction in PASE score along with a decrease in SPPB and NEADL scores, and that physical activity was positively associated with activities of daily living. However, physical activity was not related to physical performance in this sample in advanced age. The type of physical activities (housework vs. recreational activities) could have masked this association. Interestingly, the current study found that participants with a better socio-economic status had poorer functional and physical performance than those with worse socio-economic status, challenging the wealth–health directionality [63]. Socio-economic status and food insecurity are factors affecting inadequate nutrient intake that has been discussed throughout the literature [14]. A previous study of 70-year-old Korean people reported total weight-adjusted protein intake increased proportionately with household income and education status [64]. Our study on nonagenarians found the contrary. This prompts the question of whether the socio-economic status of people in advanced age plays a role in the quality and quantity of protein consumed and the further impact on physical performance capability. Further research is needed to understand the interplay between socio-economic status, living environment, and health status impacting dietary intake in older adults.

### 4.4. Strengths and Limitations

This study is part of the LiLACS-NZ cohort study, allowing us to draw associations between past protein intake and functional status over ten years. Protein intake was recorded in a standardised way over two time points, ensuring continuity for comparison. This allows us to gain insight into the prevalence of inadequate protein intake and the differences in protein quantity and quality (protein sources) in this population over ten years. In addition, this prospective study highlighted the temporal sequence of events. It enabled the inclusion and examination of multiple exposure variables and potential confounding related to the outcome of physical function (NEADL score and SPPB score).

The main limitation of this study was the relatively small sample size of 81 participants, reducing the ability to produce ethnic-specific analyses. The annual mortality rate in the first four years of the follow-up for the LiLACS-NZ study was approximately 9%, and this was the main reason for attrition [65]. Due to this population being of advanced age, it is unsurprising that many participants were lost to follow-up or passed away before the 10-year study interviews. The small sample may introduce type II errors, and we advise a cautious interpretation of the results. Health conditions and the use of medication may also affect the study outcome. At baseline, 93% of the sample had two or more chronic conditions; the median of health conditions and use of prescribed medications was five, respectively [65,66]. The ubiquitous multimorbidity in the sample restricts our ability to untangle the cause–effect relationship between physical activity and chronic conditions. Considering the issue of over-adjustment, we selected physical activity level as a covariate in the model based on the notion that nutrition status and physical activity go hand in hand.

We completed Intake24 (an electronic version of the original interviewer-led 24 h MPR 24-h) once to reduce the participant burden in the current study. Completing one MPR may be less accurate, but it is common practice in research studies when balancing practicality, resources, and data [67]. MPR has been validated and shown to produce similar estimates of group intake in those who do not have extreme intake [68]. MPR can be less accurate when interviewing people with cognitive decline as it relies heavily on memory; thus, we aimed to mitigate this by having a Kaiāwhina or support person with participants to assist. Intake24, with the additions of New Zealand foods from the FOODfiles 2016 database, has not been validated in a New Zealand context. Intake24 underestimated energy intake by 1%, and mean macro- and micronutrient intake was within 4% of those recorded in interviewer-led recalls in a UK population [40].

To our knowledge, this is the first cohort study to assess protein intake in the nonagenarian age group. These findings fill a gap in the existing literature. Causality cannot be proven. Therefore, we can only show an association and recommend a cautious interpretation of the findings.

The current study’s findings may improve quality of life of adults of advanced age in the future by supporting international recommendations for adequate protein intake to support physical function in advanced age.

Future research on the effect of protein from plant and animal sources and their impact on physical function over time is required to confirm these findings. Exploration of Indigenous versus Westernised protein food sources and their impact on physical function in needed. Insights from the current study will contribute to the design of future prospective studies and trials of supplements in very old adult populations.

## 5. Conclusions

The LiLACS-NZ cohort study highlighted a reduction in protein intake with a change in protein sources from animal to plant-based protein over ten years in octogenarians. The study found protein intake 10 years prior was associated with functional status ten years later. The novel nature of this research was that it documents change in dietary intake in nonagenarians related to physical function over time.

## Figures and Tables

**Figure 1 nutrients-15-01664-f001:**
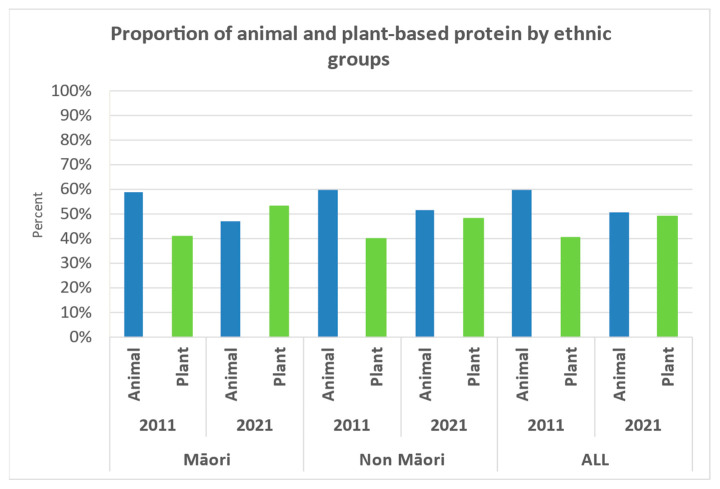
The proportion of animal and plant-based protein in 2011 and 2021 by ethnic groups.

**Figure 2 nutrients-15-01664-f002:**
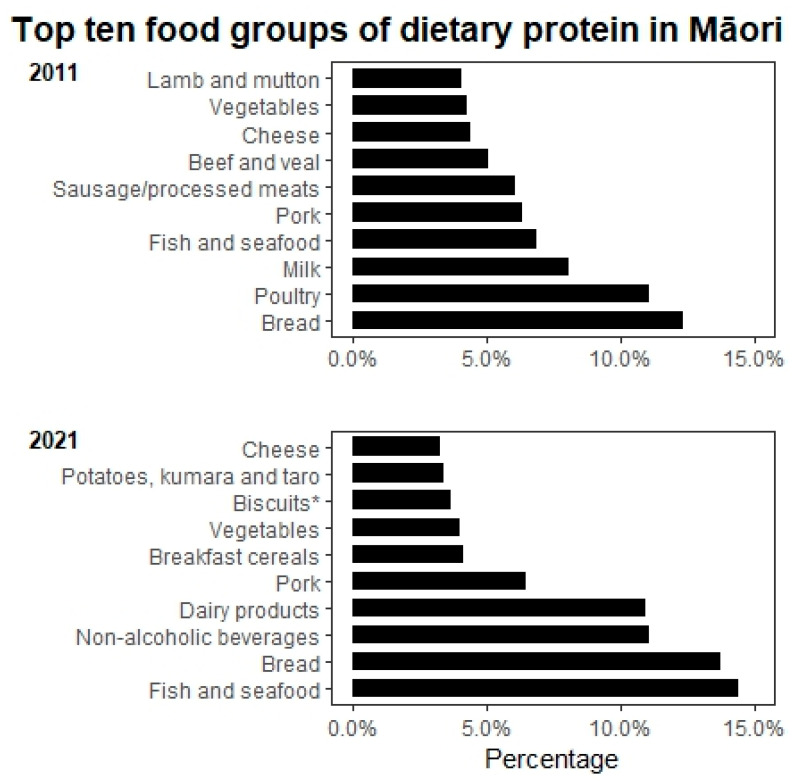
Top ten food groups of dietary protein intake for Māori in 2011 and 2021. Notes: Classification of food groups were based on the 2008/09 New Zealand Adult Nutrition Survey NZ food groups. * Sweet biscuits (plain, chocolate coated, fruit filled, cream filled), crackers.

**Figure 3 nutrients-15-01664-f003:**
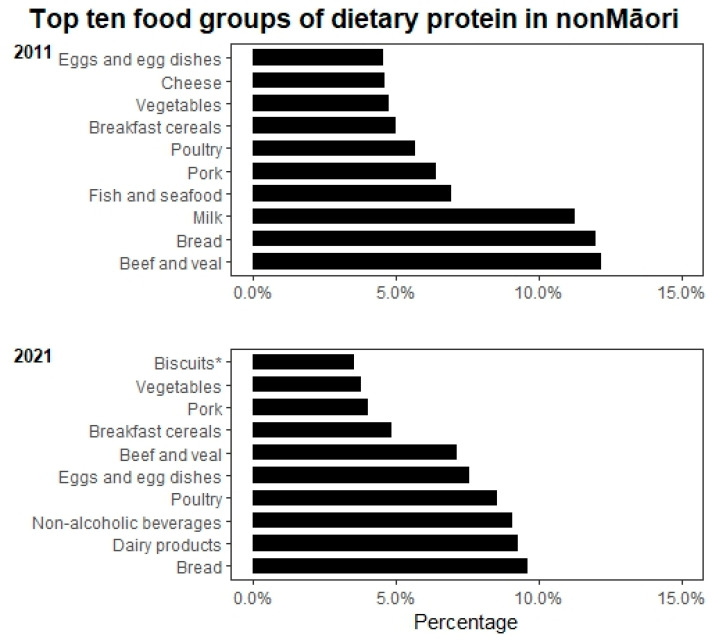
Top ten food groups of dietary protein intake for non-Māori in 2011 and 2021. Notes: Classification of food groups were based on the 2008/09 New Zealand Adult Nutrition Survey NZ food groups. * Sweet biscuits (plain, chocolate coated, fruit filled, cream filled), crackers.

## Data Availability

We are happy to share LiLACS NZ data with interested researchers on condition of a mutually acceptable agreement in data usage. The process to apply for LiLACS NZ data access is available at [https://www.fmhs.auckland.ac.nz/en/faculty/lilacs.html] (accessed on 28 March 2023).

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
