# Peer review of "Dietary Protein Intake and Physical Function in Māori and Non-Māori Adults of Advanced Age in New Zealand: LiLACS NZ"

_nutrients, 2023, doi:10.3390/nu15071664_

Round 1

Reviewer 1 Report

Dear authors,

You performed an interesting study on protein intake in the past and its relation to physical performance 10 years later. I have some concerns related to this study.

A ten year period is rather long and the sample size is (too) small. There is no information on for example illness, use of medication, etc. that may affect your outcome. It is not clear what statistic analyses were performed on these data. You mention several in the method section, but I cannot see this differentiation in the results section.

The number of Maori participants is too low to draw conclusions, in my opinion. I also think it is not correct to use means and SD's in these low numbers. The table also shows large SD's, probable affecting outcomes. 

Please provide a complete title for Figure 1 (I think it is related to Maori participants)

Author Response

Dear Reviewer 1, 

Thank you for your inputs. Please see the attachment of our point-by-point response to your comments.

Ngaa mihi

Reviewer 2 Report

Comments:

Review of the paper

nutrients-2267512

This study aimed to investigate dietary protein intake and physical function in Māori and non-Māori adults of advanced age in New Zealand. The study included Eighty-one participants (23 Māori and 54 Non-Māori). The results of the present study are very interesting, it is helpful for improving the quality of life of adults of advanced age in the future by supporting international recommendations for adequate protein intake to support physical function in advanced age. The novel nature of this research was that it documents change in dietary intake in nonagenarians related to physical function over time. The manuscript is well written, and the results are well discussed. However, several issues should be addressed before publication.

INTRODUCTION

- The article should supplement the references in recent years to prove the prevalence of protein intake of the elderly.

MATERIALS AND METHODS

-Too few follow-up visits are an obvious limitation of this study, which should be considered and explained in the results.

RESULTS

-The changes in Cholesterol intake and Fat intake are also statistically significant, so they should be mentioned.

-As for the top ten groups of dietary protein, the authors had better add some description.

FIGURE

- The figures were not well prepared, authors should use more professional drawing software to prepare the figure, instead of excel. 

Author Response

Dear Reviewer 2, 

Thank you for your inputs. Please see the attachment of our point-by-point response to your comments.

Ngaa mihi

Round 2

Reviewer 1 Report

Dear authors,

Thank you for this revised manuscript. I still think the sample size is low to perform statistical analyses, but I appreciate that this is now emphasized more in the manuscript as a limitation of the study. 

Author Response

Dear Reviewer, 

Thank you for your effort in improving the clarity of the paper